# Oral Trehalose Intake Modulates the Microbiota–Gut–Brain Axis and Is Neuroprotective in a Synucleinopathy Mouse Model

**DOI:** 10.3390/nu16193309

**Published:** 2024-09-30

**Authors:** Solène Pradeloux, Katherine Coulombe, Alexandre Jules Kennang Ouamba, Amandine Isenbrandt, Frédéric Calon, Denis Roy, Denis Soulet

**Affiliations:** 1Centre de Recherche du CHU de Québec, Québec, QC G1V 4G2, Canada; solene.pradeloux@crchudequebec.ulaval.ca (S.P.); katherine.coulombe@vrr.ulaval.ca (K.C.); alexandre-jules.kennang-ouamba.1@ulaval.ca (A.J.K.O.); amandine.isenbrandt@crchudequebec.ulaval.ca (A.I.); frederic.calon@crchudequebec.ulaval.ca (F.C.); denis.roy@fsaa.ulaval.ca (D.R.); 2Faculté de Pharmacie, Université Laval, Québec, QC G1V 0A6, Canada; 3Faculté des Sciences de l’Agriculture et de l’Alimentation, Université Laval, Québec, QC G1V 0A6, Canada

**Keywords:** Parkinson’s disease, trehalose, gut microbiota, microbiota–gut–brain axis, PrP-A53T, GLP-1

## Abstract

Parkinson’s disease (PD) is a neurodegenerative disease affecting dopaminergic neurons in the nigrostriatal and gastrointestinal tracts, causing both motor and non-motor symptoms. This study examined the neuroprotective effects of trehalose. This sugar is confined in the gut due to the absence of transporters, so we hypothesized that trehalose might exert neuroprotective effects on PD through its action on the gut microbiota. We used a transgenic mouse model of PD (PrP-A53T G2-3) overexpressing human α-synuclein and developing GI dysfunctions. Mice were given water with trehalose, maltose, or sucrose (2% *w*/*v*) for 6.5 m. Trehalose administration prevented a reduction in tyrosine hydroxylase immunoreactivity in the substantia nigra (−25%), striatum (−38%), and gut (−18%) in PrP-A53T mice. It also modulated the gut microbiota, reducing the loss of diversity seen in PrP-A53T mice and promoting bacteria negatively correlated with PD in patients. Additionally, trehalose treatment increased the intestinal secretion of glucagon-like peptide 1 (GLP-1) by 29%. Maltose and sucrose, which break down into glucose, did not show neuroprotective effects, suggesting glucose is not involved in trehalose-mediated neuroprotection. Since trehalose is unlikely to cross the intestinal barrier at the given dose, the results suggest its effects are mediated indirectly through the gut microbiota and GLP-1.

## 1. Introduction

Parkinson’s disease (PD) is the second-most common age-related neurodegenerative disorder, with 8.5 million patients reported worldwide in 2019 [1]. It is a progressive neurological condition characterized by the loss of dopaminergic neurons in the substantia nigra pars compacta (SNpc) [2]. It is also a synucleinopathy, with sometimes presence in the nigrostriatal region of neuronal inclusions. Those are named Lewy bodies (LB) and are mainly composed of the misfolded protein alphα-synuclein (α-syn) [3], promoting neuronal apoptosis [4,5]. The neurodegeneration reduces the supply of dopamine (DA) to the striatal region, causing the development of motor symptoms characteristic of the disease, such as tremors and bradykinesia [2]. Non-motor symptoms, including gastrointestinal disturbances, precede the onset of motor symptoms, highlighting the potential involvement of the gut in PD progression [6]. Patients with PD also show reduced levels of DA in the gut [7], suggesting dysfunction of the enteric dopaminergic system, and present an alteration of the intestinal microbiota [5,8,9,10], such as specific bacterial abundance changes and reduced production of short-chain fatty acids (SCFAs) [8,10,11,12].

According to Braak’s hypothesis, the first α-syn aggregates appear in the enteric nervous system (ENS) and could spread from the intestinal region to the brain, suggesting the involvement of the gastrointestinal system and the gut microbiota in the progression of the disease via the microbiota–gut–brain axis [13]. A dysbiosis of the gut microbiota, as observed in PD patients, can increase intestinal permeability and trigger an immune response, affecting the host’s metabolism [14]. Activation of enteric glial cells may contribute to the initiation of α-syn misfolding [15]. The role of the gut microbiota in communication within the gut–brain axis in healthy and disease conditions has recently been recognized. Also, several bacterial strains can modify levels of neurotransmitter precursors in the gut lumen and even modulate the synthesis of several neurotransmitters, including serotonin (5-HT) and DA [16,17]. Thus, the gut microbiota can interact with the central nervous system (CNS) bidirectionally via the vagus nerve [18] or via metabolites and endotoxin translocation from the lumen to the circulation [17]. It has been proposed that the consumption of polyunsaturated fatty acids is beneficial to the brain [19]. SCFAs are essential for intestinal homeostasis, contributing to the maintenance of the intestinal barrier integrity [20], neuroplasticity, and brain function and behavior [21]. SCFAs affect brain neurochemistry; they modulate the expression levels of tryptophan 5-hydroxylase 1, the enzyme involved in the synthesis of 5-HT and tyrosine hydroxylase (TH), which is a rate-limiting enzyme in the biosynthesis of DA, noradrenaline and adrenaline [22,23].

To this day, PD is incurable; the available pharmacological treatments can only attenuate motor dysfunctions with no effect on disease progression [24]. It is, therefore, essential to seek alternative solutions to protect dopaminergic neurons. Several studies have shown the importance of diet on the development of PD and the potential of nutraceutical treatments [25,26]. In this regard, trehalose (O-α-D-glucopyranosyl-[1→1]-α-D-glucopyranoside) is a disaccharide composed of two molecules of glucose, found primarily in plants, fungi, archaea, bacteria, and insects [27]. The available literature presents some evidence of neuroprotective abilities related to trehalose intake in models of PD [28,29]. Oral administration of trehalose has been shown to improve motor function and survival in animals modeling proteinopathies, reduce astrogliosis in Parkin knock-out (KO) mice, and protect against striatal dopamine loss in the MPTP (1-methyl-4-phenyl-1,2,3,6-tetrahydropyridine) model of PD [28,29,30]. Interestingly, trehalose would be neuroprotective only when administered orally in animals [30]. However, the mechanisms of action surrounding this neuronal protection are little or not identified. The mechanism of transport of trehalose through the different physiological barriers is still poorly understood, and it is necessary to mention that vertebrates do not have the capacity to produce trehalose endogenously or to transport it actively since they do not have the intestinal membrane transporters needed [27,30]. A direct action of trehalose on the neurons of the nigrostriatal pathway is therefore unlikely. Nevertheless, trehalose metabolites (glucose) or indirect effects via the gut microbiota could be involved. Indeed, several studies demonstrated the ability of trehalose to promote the proliferation of specific gut bacteria [31,32]. Trehalose can be cleaved into two molecules of glucose by the bacterial and mammalian intestinal enzyme trehalase.

The present study focuses on investigating the neuroprotective effects of trehalose intake and identifying the mechanisms involved in a transgenic murine model of PD. We hypothesize that exposure to this sugar would modulate the gut microbiota composition and activity (production of immunomodulatory SCFAs) and influence the microbiota–gut–brain axis to induce indirect neuroprotective effects. Therefore, the study aims to explore the mechanisms of action of trehalose, focusing on its action on the microbiota–gut–brain axis in a progressive mouse PD model.

## 2. Materials and Methods

### 2.1. Animals

In this study, we used a transgenic mouse model of synucleinopathy, the strain B6.Cg-2310039L15RikTg(Prnp-SNCA*A53T)23Mkle/J → Prp-A53T Line G2-3 from Jackson Laboratories (Hualphα-syn(A53T) transgenic line G2-3 Strain [33]). These transgenic mice were characterized by an accumulation of the mutated A53T form of the α-syn protein, present in certain familial forms of the disease in patients. The selected mouse strain showed an accumulation of the protein of interest and its insoluble aggregated form in several regions of the brain, including the striatum and substantia nigra [34]. Studies characterizing the Prp-A53T Line G2-3 mouse strain observed constipation in transgenic animals from the age of 3 m, i.e., several months before the onset of motor symptoms (after 9 m of age). This is the only known model that combines both gastrointestinal dysfunction and synucleinopathy.

Heterozygous PrP-A53T male mice were crossed in our facility with female mice not carrying the mutation to generate the experimental model necessary for carrying out our protocol: male mice carrying the mutation (PrP-A53T, n = 32) and their non-carrier (NC) control littermates (NC, n = 32) were used. Mice were housed in ventilated cages on a 12 h/12 h dark/light cycle at 22 °C. Animals had free access to water and appropriate food. Mice were weighed every month. All animal husbandry and experiments were approved by the animal research committee of the Centre de recherche du CHU de Québec—Université Laval and performed according to the Canadian Guide for the Care and Use of Laboratory Animals (Protocol number: CPAUL-3 20-609). All efforts were made to minimize animal suffering and to reduce the number of mice used. Mice were monitored twice a day for the duration of the protocol for the following signs: infections, injuries, dehydration, malnutrition, hypo- or hyperthermia. The human endpoint established based on the University Animal Care Committee (CUPA) guidelines for Laval University Animal Care Committees (CPAU) (PNF ETH-10) were: irreversible or prolonged inability of mice to eat or drink; prolonged lateral recumbency; severe pain or suffering that cannot be relieved by analgesic agents or other treatments; convulsions; skin lesions affecting more than 20% of the body surface area; severe or prolonged untreatable dehydration; severe or prolonged breathing difficulties; emaciation; severe uncontrolled bleeding; hyperthermia or severe hypothermia; lethargy, unconsciousness, shock; severe paralysis; rapid weight loss.

### 2.2. Groups and Treatments

Animals were randomly allocated to four groups, with eight mice in each group. Mice received a control (water-only), trehalose (T0167, Sigma-Aldrich, Oakville, ON, Canada), maltose (M5885, Sigma Aldrich, Oakville, ON, Canada), or sucrose (573113, MilliporeSigma, Oakville, ON, Canada) diluted in water at 2% *w*/*v* disaccharide for a 6.5-m period from the age of 3 m. Mice drank around 10 mL of water per day. Supplemented water was administered in bottles and changed every two days. The dose of trehalose, maltose, or sucrose was approximatively 3.3 g/kg/day per animal. The dose of trehalose used was selected from published reports showing trehalose neuroprotective effects [29,35]. Maltose and sucrose were used as control disaccharides for trehalose.

### 2.3. In Vivo Tests

#### 2.3.1. Open Field Test

Open field tests were performed at the following time points, 3, 6, 7, 8 and 9 m, for all mice (Figure 1) to analyze the progressive locomotor dysfunction and anxiety-like behavior, as previously described [36,37]. Mice were allowed to move freely in a box of 1 m^2^ area for 30 min and were tracked using an original program written in Matlab 2018b environment (MathWorks, Natick, MA, USA) [38]. Locomotor activity was quantified as the distance traveled in 30 min and percentage of active time. Anxiety-like behavior was measured as the percentage of time in the inner zone.

At the end of the test, feces were collected in Eppendorf tubes for further measurements. Feces were counted and weighed to obtain the wet mass; then the tubes containing the stools were placed on a heating block at 60 °C overnight for the water to evaporate. Afterwards, feces were once again weighed to obtain the dry mass in order to calculate the proportion of water, consequently allowing us to obtain three indirect measures of intestinal motility.

#### 2.3.2. Stool Harvest for Microbiota Profiling

Fecal samples used for 16S RNA sequencing analyses were collected in sterile conditions at fixed times at the three-, six-, and nine-month time points (Figure 1). Mice were isolated in clean cages for 1 h, and the samples were freshly collected. If a mouse did not defecate, abdominal massage was performed to stimulate fecal excretion. About 250–300 mg of fecal matter was collected per animal. Samples were immediately placed on dry ice and then stored at −80 °C.

#### 2.3.3. Nesting Test

A nesting test was performed to measure the quality of the mice’s nests, giving an indication of their levels of anxiety and depression [39,40]. Mice were isolated in individual cages four days before euthanasia. Mice had 48 h of habituation to the new environment; then a new nestlet material was put down in the cage, and the nests’ quality was scored following predetermined criteria at the following time points: 1, 4, 8, 12, 24, and 48 h. Scores ranged from 0 to 4 in increments of 0.5 and considered the size of the nestlet pieces, the shape, and the depth of the nest.

### 2.4. Microbiota Analysis

#### 2.4.1. DNA Extraction

DNA was recovered from the collected stools using the DNeasy^®^ PowerLyzer^®^ PowerSoil^®^ Kit (12855-100, Qiagen Germantown, MD, USA) by bead-based isolation of DNA, following the kit instructions. The recovered DNA was stored at 4 °C. Subsequently, we used an Infinite^®^ 200 PRO (Tecan US, Inc., Morrisville, NC, USA) to analyze the concentration and purity of the extracted DNA. Then, all samples were diluted with RNase Free water to reach a concentration of 10 ng/µL and then sent to Université Laval genomic core facility (IBIS, Université Laval, Quebec City, QC, Canada) to perform the 16S rRNA gene amplicon sequencing.

Briefly, amplification of the 16S rRNA gene V3-V4 region was performed using the sequence-specific regions described in Klindworth et al. [41] using a two-step dual-indexed PCR approach specifically designed for Illumina instruments. In a first step, the gene-specific sequence was fused to the Illumina TruSeq sequencing primers (Table 1), and PCR was carried out in a total volume of 50 µL that contained 1× Q5 buffer (B9027S, New England Biolabs, Ipswich, MA, USA), 0.25 µM of each primer, 200 µM of each dNTPs, 1 U of Q5 High-Fidelity DNA polymerase (M0491L, New England Biolabs, Ipswich, MA, USA), and 1 µL of template cDNA. The PCR started with an initial denaturation at 98 °C for 30 s, followed by 35 cycles of denaturation at 98 °C for 10 s, annealing at 55 °C for 10 s, extension at 72 °C for 30 s, and a final extension at 72 °C for 2 min. The PCR reaction was purified using the sparQ PureMag PCR cleanup kit (95196-450, Quantabio, Beverly, MA, USA). The quality of the purified PCR products was checked on a 1% agarose gel. Five-to-tenfold dilution of these purified products was used as a template for a second PCR step with the goal of adding barcodes (dual-indexed) and missing sequences required for Illumina sequencing. Cycling for the second PCR was identical to the first PCR but with 12 cycles. PCR reactions were purified as above, checked for quality on a DNA7500 Bioanalyzer chip (5067-1506, Agilent, Santa Clara, CA, USA), and then quantified spectrophotometrically using a Spark 10 M plate reader (Tecan US, Inc.). Barcoded amplicons were pooled in equimolar concentration for sequencing on the Illumina Miseq using a 600-cycle kit according to the manufacturer’s instructions. Primers used in this work contain Illumina-specific sequences protected by intellectual property (Oligonucleotide sequences© 2007–2013 Illumina, Inc. (San Diego, CA, USA). All rights reserved. Derivative works created by Illumina customers are authorized for use with Illumina instruments and products only. All other uses are strictly prohibited).

#### 2.4.2. 16S rRNA Gene Sequencing and Sequence Analysis

The V3-V4 region of the 16S rRNA gene was amplified using the primer pairs 341F (5′-CCTACGGGNGGCWGCAG-3′) and 805R (5′-GACTACHVGGGTATCTAATCC-3′), and sequencing was performed on an Illumina MiSeq system. Primer removal on demultiplexed raw sequences was performed using Cutadapt (version 4.1) software [42]. Reads were then processed in the DADA2 (version 1.24.0) pipeline [43] implemented in the R (version 4.2.1) environment. Paired reads of inferred sequence variants were merged to obtain the full amplicon sequence variants (ASVs). Following chimera removal, ASVs were assigned a taxonomy based on the DADA2-formatted Silva version 138.1 reference database. The phylogenetic tree constructed with the DECIPHER (version 2.24.0) package [44] and phangorn (version 2.10.0) package [45] was combined to the ASV table, the taxonomy data, and the sample metadata to build a phyloseq object using the phyloseq (version 1.40.0) package [46].

The Chao1, Shannon, and inverse Simpson indices computed in the phyloseq package were used to describe the alpha-diversity of bacterial communities in NC and PrP-A53T mouse feces. The taxonomic profile was visualized by plotting taxa relative abundance at the family level. For beta-diversity analyses, principal component analysis (PCA) and principal coordinate analysis (PCoA) performed on centered log-ratio (CLR) and phylogenetic isomeric log-ratio (PhILR) transformed data [47,48] were used to compare the compositional and phylogenetic structures of the microbiota of NC and PrP-A53T mouse genotypes.

### 2.5. Short Chain Fatty Acids Analysis

Fecal samples were weighed and suspended in 1 mL of Milli-Q water per 100 mg of sample. The suspension was homogenized with a Bead Ruptor Elite (Omni International, Kennesaw, GA, USA) at 4 m/s for 2 min and then centrifuged for 10 min at 18,000× *g* at 4 °C. A total of 500 µL of supernatant was collected in a clean tube for the liquid-liquid extraction of SCFAs and quantified by gas chromatography coupled to a flame ionization detector, as described in Roussel et al. [49]. Supernatants were spiked with a solution containing an internal standard (4-methylvaleric acid, Sigma) and H3PO4 (Fisher, Mississauga, ON, Canada) 10% to obtain a pH of 2. A volume of methyl tert-butyl ether (Sigma), equivalent to the volume of diluted sample, was added to extract SCFAs by vortexing for 2 min. Samples were then centrifuged for 10 min at 14,000× *g* at 4 °C, and organic phases were transferred to a glass vial. SCFAs analysis was performed on a GC-FID system (Shimadzu, Kyoto, Japan), consisting of a GC2010 Plus gas chromatograph equipped with an AOC-20s auto-sampler, an AOC-20i auto-injector, and a flame ionization detector. The system was controlled by GC solution software (version 2.40). One microliter of organic phase was injected in a split mode into a Nukol capillary GC column (30 m 0.25 mm id, 0.25 mM film thickness, Supelco analytical), and hydrogen was used as carrier gas. The injector and detector were set at 250 °C. The oven temperature was initially programmed at 60 °C and then increased to 200 °C at the rate of 12 °C/min and held at this temperature for 2 min. SCFAs were quantified using a five-point calibration curve prepared with a mix of standards (acetic acid, propionic acid, butyric acid, isobutyric acid, valeric acid, and isovaleric acid, Sigma) extracted following the same procedure as samples.

### 2.6. Tissue Preparation

All animals were sacrificed at 9.5 m of age, the day following the last measure of the nesting test, by intracardiac perfusion with 0.1 M phosphate-buffered saline (PBS), performed after deep anesthesia with a ketamine and xylazine mixture (respectively, 100 mg/kg and 10 mg/kg). A drop of blood was placed on a blood glucose test strip (06453988, Accu-Check Aviva, Indianapolis, IN, USA), and blood glucose levels were read using a blood glucose meter for people with diabetes (Accu-Check Aviva). The brain was quickly removed and separated into two hemispheres. One hemisphere was frozen in isopentane in dry ice (−40 °C) for a few seconds and then placed in dry ice and stored at −80 °C. The second hemisphere was post-fixed with 4% paraformaldehyde (PFA) for 48 h and then stored in a solution composed with 20% sucrose and 0.05% sodium azide in 0.1 M PBS pH 7.4 at 4 °C. The intestine was recovered and post-fixed in 4% PFA for 48 h and then stored in a solution composed of 20% sucrose and 0.05% sodium azide in 0.1 M PBS pH 7.4 at 4 °C.

Fixed brains were frozen in a dry ice and ethanol mixture, mounted on a microtome (Leica Microsystems Inc., Richmond Hill, ON, Canada), and cut into 30 μm thick coronal sections. Collected brain sections were immersed in a tissue cryoprotectant solution (0.05 M PBS pH 7.3, 30% ethylene glycol, and 20% glycerol) and stored at −20 °C until used for immunofluorescence. Five to six sections (1–10 mm^2^/section) of small intestine sampled in the distal ileum were microdissected to reveal the myenteric plexus hidden between muscle layers [50].

### 2.7. Immunofluorescence Analysis

Immunoreactivity quantifications and counts were performed by immunofluorescence. Myenteric plexus (5–6 sections) and brain sections containing the striatum (4–5 sections) or the SNpc (5–6 sections) were incubated for 1 h at 100 °C in sodium citrate for antigen retrieval before a 30-min blocking treatment with a solution of 0.4% Triton X-100 and 5% donkey serum (Sigma-Aldrich, Oakville, ON, Canada) in PBS 1×. Free-floating tissues were stained overnight with different primary antibodies, followed by a 2-h incubation with secondary antibodies in a blocking solution. Nuclear counterstaining with 0.022% DAPI (Invitrogen Corporation, Waltham, MA, USA) was performed before mounting sections. See Table 2 for antibody listing.

Marked sections were imaged using a Zeiss AxioScan.Z1 Digital Slide Scanner and Zen 2.3 acquisition software (Carl Zeiss, Toronto, ON, Canada). Microdissected myenteric plexus sections and brain sections containing the SNpc were imaged per animal with a 20× objective lens (NA 0.45). Brain sections of the striatum were imaged with a 10× objective lens (NA 0.45). Mean pixel intensity was determined on images for immunoreactivity quantification using MathWorks Matlab^®^ 2018a software, and for density count, labeled cells were calculated as the number of positive cells per area (mm^2^) using ImageJ Fiji software (ImageJ 1.54f Java 1.8.0_322). Mean values were calculated with the microdissected sections for each animal. All images were captured blindly, as were undertaken in the data analyses [51].

### 2.8. Statistical Analysis

Data were evaluated with the Shapiro–Wilk test to determine normality; outliers were discarded using the interquartile range method. Since data were normally distributed, group comparisons were conducted using a two-way analysis of variance (ANOVA) with genotype and treatment as independent variables, followed by Tukey’s post-hoc tests. Longitudinal comparisons between different time points for animals with the same genotype and treatment were performed using a one-way ANOVA, also followed by Tukey’s post-hoc tests. Results for each animal are represented as the mean ± SEM of six to eight mice per group. Results are considered statistically significant if *p* < 0.05. All statistical analyses were performed with Prism 9 (GraphPad Software Inc., La Jolla, CA, USA) software.

For the microbiota analyses, the Wilcoxon rank test was used to access the similarity among bacterial communities of NCs and PrP-A53T, and statistical significance was established for *p* < 0.05. Linear discriminant analysis effect size (LEfSe) algorithm [52] was applied to identify taxa that explained differences between bacterial communities. The functional metagenomic composition of NC and PrP-A53T bacterial communities was predicted by utilizing the PICRUSt2 software v2.5.3 [53]. LEfSe analysis was performed to find differentially abundant functional pathways inferred from both communities. The taxa contribution of predicted pathways that were differentially abundant between the two genotypes was determined and visualized using alluvial plots.

## 3. Results

### 3.1. Effect of Genotype and Sugar-Enriched Diet on Mouse Weights and Glycemia

Mice were weighed every month, revealing a progressive weight gain of 19 g for the NC mice between 3 and 9.5 m, representing a 67% increase in body mass (NC mice age factor F (7, 219) = 95.21, *p* < 0.0001; Figure 2A). PrP-A53T mice had a lower weight gain (+6 g) compared to NC mice, representing a 21% increase in body mass (PrP-A53T mice age factor F (7, 214) = 0.3678, *p* = 0.0489), and they had a significantly lower weight than the NCs at the ages of 7 m (10.6 g, *p* = 0.0033), 8 m (11.8 g, *p* = 0.0014), 9 m (13.6 g, *p* = 0.0037), and 9.5 m (13.3 g, *p* = 0.046). Drinking of trehalose, maltose, and sucrose did not affect the weights of mice.

PrP-A53T mice also showed a 34% lower glycemia at the time of euthanasia compared to NC mice (genotype factor F (1, 53) = 42.29, *p* < 0.0001; Figure 2B). Intake of maltose and sucrose did not impact glycemia levels in NC mice (*p* > 0.9999 for both maltose and sucrose) or in PrP-A53T mice (respectively, *p* > 0.9999 and *p* = 0.9853). However, NC mice treated with trehalose had a lower glycemia at euthanasia than the other NC mice (*p* = 0.0263). This lowered glycemia was not observed in PrP-A53T mice whatever the treatment (*p* = 0.9979). Thus, it appears that trehalose has a hypoglycemic effect, but only in NC mice.

### 3.2. Effect of Genotype and Sugar-Enriched Diet on Motor and Non-Motor Symptoms

The results of the open field tests (Figure 3A–C) revealed an important increase in PrP-A53T mice motor activity with aging, and PrP-A53T mice moved more often than NC mice at 6 m (*p* < 0.01), 7 m (*p* < 0.01), 8 m (*p* < 0.0001), and 9 m of age (*p* < 0.01). Thus, PrP-A53T mice were more active than NC mice from 6 m of age, and the administration of neither trehalose, maltose, nor sucrose altered their motor activity.

Anxiety and depressive-like behaviors were measured by the time mice spent in the inner zone of the open field (Figure 3D) and using the nesting test to score the quality of nest (Figure 4). An anxiety onset was observed from 6 m of age (*p* < 0.01) with less time spent in the open field inner zone than at 4 m of age. No anxiety-like behavior was observed in NC mice (Figure 3D). Nesting building scores at 9.5 m of age showed that NC mice were actively building their nest over a 48-h period, with a score progressing from 0.69 at 1 h to 2.9 at 48 h (time factor F (5, 168) = 51.57, *p* < 0.0001; Figure 4B). On the other hand, PrP-A53T mice built little or no nest, with scores (0.64 ± 0.08) significantly lower compared to NC mice (2.84 ± 0.17, *p* < 0.001) after 48 h of nesting building test, indicating the presence of anxiety and depression. Overall, these behavioral results suggest that the main effect of α-syn overexpression is development of hyperactivity and anxiety and depressive-like behavior in mice, beyond what is normally observed with age. Those behaviors were not impacted by the consumption of trehalose, maltose, or sucrose in NC mice (treatment factor F (3, 28) = 2.228, *p* = 0.1069) and in PrP-A53T mice (treatment factor F (3, 24) = 1.545, *p* = 0.2287).

The results obtained from the analysis of intestinal motility showed that the number of feces excreted in 30 min (genotype factor F (1, 48) = 15.91, *p* = 0.0002; Figure 5A), their weight (genotype factor F (1, 48) = 21.57, *p* < 0.0001; Figure 5B), and the proportion of water in the stools (genotype factor F (1, 47) = 24.50, *p* < 0.0001; Figure 5C) were lower in PrP-A53T mice than NC mice at 9.5 m old. Thus, PrP-A53T mice seem to have a less active intestinal transit compared to NC mice. The intake of trehalose (*p* = 0.9997), maltose (*p* > 0.9999), or sucrose (*p* = 0.9446) did not significantly impact the intestinal transit in NC mice. The intake of trehalose (*p* = 0.9998), maltose (*p* = 0.9955), or sucrose (*p* > 0.9999) did not significantly impact intestinal transit in PrP-A53T mice.

### 3.3. Accumulation of Human α-Syn in PrP-A53T Mice

To confirm the expression and accumulation of human α-syn (h-α-syn) in the PrP-A53T mouse model, we analyzed the immunoreactivity of a specific phosphorylated serine 129 (p-S129) h-α-syn antibody in the SNpc and striatum of all mice (Figure 6). As expected, fluorescence imaging confirmed the absence of p-S129 h-α-syn in the brain of NC mice (background signal around 20,000 A.U. in the SNpc and 2400 A.U. in the striatum) and the presence of the protein in the PrP-A53T mice, in the SNpc (genotype factor F (1, 51) = 239.5, *p* < 0.0001; Figure 6A,C) and the striatum (genotype factor F (1, 53) = 601.0, *p* < 0.0001; Figure 6B,D). Furthermore, no significant effect was observed for any treatments in PrP-A53T mice (treatment factor F (3, 51) = 1.349, *p* = 0.2689 in the SNpc; F (3, 53) = 1.240, *p* = 0.3043 in the striatum), suggesting that trehalose and other sugars do not regulate accumulation and phosphorylation of h-α-syn in mice (water vs. trehalose in SNpc, *p* = 0.0624; water vs. trehalose in striatum, *p* = 0.2762). To conclude, these results confirm the presence of p-S129 h-α-syn in the CNS of PrP-A53T mice and that trehalose treatment does not modulate h-α-syn expression and phosphorylation.

### 3.4. Neurodegeneration in PrP-A53T Mice and Neuroprotective Effect of Trehalose

We investigated the integrity of both the nigrostriatal pathway and the myenteric plexus and the impact of trehalose intake using TH, a rate-limiting enzyme in the synthesis of DA, as a marker for DA neurons. The distribution of TH labeling is representative of what is expected from labeling of DA neurons and fibers, confirming the specificity of the antibody used. Quantification of the mean intensity of TH in SNpc revealed a 25% loss of nigral TH in PrP-A53T mice treated with water, maltose, and sucrose (genotype factor F (1, 52) = 9.197, *p* = 0.0038; Figure 7A,B). This loss was prevented by the trehalose treatment (*p* = 0.0014). TH labeling for the DA projections in the striatum showed a 38% loss of TH projections in PrP-A53T mice treated with water, maltose, and sucrose (genotype factor F (1, 50) = 24.38, *p* < 0.0001; Figure 7C,D), while the trehalose treatment prevented the loss of DA projections (*p* = 0.0018).

Catecholaminergic fibers were quantified to assess neurodegeneration in the myenteric plexus. Quantification of TH^+^ fibers revealed lower levels of TH (18%) in the myenteric plexus of PrP-A53T mice treated with water, maltose, and sucrose compared with NC littermates (genotype factor F (1, 45) = 21.46, *p* < 0.0001; Figure 7E,F). The PrP-A53T mice fed with trehalose showed higher levels of TH in the myenteric plexus compared to mice receiving control water, maltose, and sucrose (respectively, *p* = 0.0027, *p* = 0.0462, and *p* = 0.0430). Cholinergic neurons were also reduced in PrP-A53T mice compared to NC mice (genotype factor F (1, 23) = 6.792, *p* = 0.0158; Appendix A), which was not impacted by trehalose treatment.

### 3.5. Effect of Genotype and Sugar-Enriched Diet on the Structure and Composition of the Mice Gut Microbiota

We compared the taxonomic profiles resulting from the 16S rRNA gene sequencing of V3-V4 regions in stool samples of mice at 9.5 m of age. PCA of the gut microbiota showed a difference in beta diversity between the NC and PrP-A53T mice receiving control water at the genus and family levels (*p* < 0.001; Figure 8A). NC mice treated with trehalose and maltose had an average beta-diversity index significantly lower compared to NC mice receiving control water (*p* < 0.05; Figure 8B) at the genus level but not at the family level, while sucrose impacted the beta diversity at the family (*p* < 0.05) but not at the genus level compared to control water-treated mice. PrP-A53T mice receiving trehalose, maltose, and sucrose also presented an average beta-diversity index significantly lower than control water-treated mice at the genus and family levels (*p* < 0.05; Figure 8C). Moreover, the beta-diversity index in PrP-A53T mice fed with maltose and sucrose was lower than that of mice fed with trehalose (*p* < 0.05). Thus, genotype modified the composition and structure of the gut microbiota at the genus and family levels. Besides, the consumption of trehalose also impacted the composition and structure of the mice’s gut microbiota, and the effect was more pronounced in PrP-A53T mice.

The analysis of differentially abundant taxa between NC and PrP-A53T bacterial communities of mice receiving plain water revealed 68 differentially abundant taxonomic clades with a linear discriminant analysis (LDA) score higher than 2.0 (Figure 9A). It showed that taxa identified as *Bacilli*, *Erysipelotrichaceae*, and *Lactobacillaceae* were significantly more abundant in NC mice’s gut microbiota, while in PrP-A53T mice, *Bacteroidaceae*, *Bacteroides*, *Defluviltaleaceae,* and *Lachnoclostridium* were significantly more abundant. For the NC mice, genera *Lachnoclostridium* and *Oscillibacter* appeared as markers of trehalose-treated mice, as were *Actinobacteriota* and *Oscillospiraceae* for sucrose-treated mice or *Lachnospiraceae*, *Eubacterium*, *Acetatifactor*, *Intestinimonas*, and *Anaerotruncus* for maltose-treated mice (Figure 9C). For PrP-A53T mice, those treated with trehalose showed significant increases in species of the family *Lachnospiraceae*, while unclassified members of the *Lachnospiraceae* (NK4A136 group) and taxa of the family *Erysipelotrichaceae* were significantly more abundant in sucrose- and maltose-treated mice, respectively (Figure 9D).

The differential abundance of predicted gut microbiota functional pathways of water- and trehalose-treated mice evaluated using LEfSe analysis on both genotypes revealed 27 functional markers associated with trehalose treatment of PrP-A53T mice (Appendix A). Among pathway classes significantly enriched were carbohydrate degradation, amino acid biosynthesis, amino acid degradation, carboxylate degradation, amine and polyamine biosynthesis, amine and polyamine degradation, and hexitol fermentation to lactate, formate, ethanol, and acetate. The most abundant taxa that contributed to the predicted pathways included *Lachnospiraceae*, *Oscillospiraceae*, *Lactobacillaceae*, *Ruminococcaceae*, and *Butyricicoccaceae* (Figure 10A). As illustrated, the *Lachnospiraceae*, which constituted the most numerically abundant group, strongly participate in hexitol fermentation to lactate, formate, ethanol, and acetate; carboxylate degradation and fermentation pathways, which then contribute to acetyl-CoA fermentation to butanoate II; L-lysine fermentation to acetate and butanoate; pyruvate fermentation to acetate and lactate II; and TCA cycle VII (acetate-producers), confirming that the *Lachnospiraceae* are involved in SCFAs biosynthesis pathway. The major SCFAs found in the stool samples at 6 m and 9.5 m were acetate (Figure 10B), propionate (Figure 10C), butyrate (Appendix A), isobutyrate (Figure 10D), valerate (Appendix A), and isovalerate (Appendix A). We observed a loss of acetate (age factor F (1, 93) = 27.08, *p* < 0.0001), propionate (age factor F (1, 95) = 14.30, *p* = 0.0003), and isobutyrate (age factor F (1, 95) = 27.90, *p* < 0.0001) between the ages of 6 and 9.5 m in the NCs as well as in the PrP-A53T mice. We also observed significantly lower levels of acetate (genotype factor F (1, 46) = 30.94, *p* < 0.0001), propionate (genotype factor F (1, 56) = 10.56, *p* = 0.002), and isobutyrate (genotype factor F (1, 46) = 31.03, *p* < 0.0001) in the fecal samples of PrP-A53T mice than NC mice at 6 m of age. Similar results were observed at 9.5 m (genotype factor for acetate F (1, 49) = 28.47, *p* < 0.0001; genotype factor for propionate F (1, 51) = 10.87, *p* = 0.0018; genotype factor for isobutyrate F (1, 48) = 7.040, *p* = 0.0108). Trehalose, maltose, and sucrose did not significantly modulate those fecal SCFAs levels in mice at 6 m or 9.5 m. Therefore, the genotype impacted the quantity of SCFAs produced in mice, but we did not observe any effect of trehalose on the levels of SCFAs at the time points studied.

### 3.6. Trehalose Restores Enteric Glucagon-like Peptide 1 (GLP-1) Immunoreactivity

Given the observed hypoglycemic effect of trehalose in Figure 2B, we wanted to investigate enteric GLP-1 abundance since it is an incretin secreted in the gut in response to the presence of glucose in the intestinal lumen. Therefore, we performed an immunofluorescence on GLP-1 in the myenteric plexus (Figure 11A). The distribution of GLP-1 immunoreactivity is consistent with what is expected of this type of labeling, confirming the specificity of the antibody used [54]. Immunoreactivity analysis revealed a significantly lower level of GLP-1 (22%) in PrP-A53T mice compared with NC littermates (genotype factor F (1, 48) = 24.84, *p* < 0.0001; Figure 11B). The intake of trehalose increased the concentration of GLP-1 in PrP-A53T mice compared with those maintained on plain water (*p* = 0.0061). These results show that the PrP-A53T genotype reduces the production of GLP-1 and that trehalose treatment restores its levels.

## 4. Discussion

In the present study, we used a progressive synucleinopathy mouse model to investigate trehalose’s benefits and mechanisms of action. For this purpose, PrP-A53T mice Line G2-3, overexpressing h-α-syn mainly in the brain and spinal cord, were compared with their NC littermates. We aimed to evaluate trehalose regimen’s impact on the microbiota–gut–brain axis and its modulatory effects on α-syn pathology. Trehalose treatment prevented the loss of mesencephalic and enteric TH^+^ neurons, indicating potential neuroprotective mechanisms (Figure 12).

Given the absence of an in vitro PD model for the enteric nervous system, the PrP-A53T Line G2-3 model was chosen for its relevance to synucleinopathy and because it is the only known model developing gastrointestinal dysfunctions [34,55]. It is worth mentioning that the various lines of PrP-A53T mice have different disease progression, symptoms, and pathological hallmarks. As such, the present model is not comparable to PrP-A53T M83 mice. Control disaccharides, maltose and sucrose, validated the neuroprotective effects’ specificity of trehalose since they did not affect the evaluated parameters. Sucrose is a glucose-fructose disaccharide with a 1→1 sugar bond. Trehalose and maltose have a similar chemical structure, both being composed of two molecules of glucose, but with a different osidic bond: 1→4 for maltose compared to 1→1 for trehalose [35,56]. Consequently, they contain the same amount of energy from a nutritional point of view. This enables us to assert that the observed effects of trehalose are not due to any disaccharide ingestion or an implication of glucose alone and to focus on trehalose for the discussion of the results. The trehalose dose selection (2% *w*/*v*, i.e., 3.3 g/kg in drinking water per day) considered previous pharmacokinetic data presented by Howson and colleagues [35], where they tested different doses of trehalose (0.89 g/kg t.i.d. or 2.67 g/kg s.i.d. in rats; 2.67 g/kg or 5.63 g/kg s.i.d. in macaques) in rodents and monkeys by oral gavage. Trehalose is hydrolyzed into two glucose molecules in the small intestine by the enzyme trehalase, expressed in the intestinal villi membrane of mammals [57]. However, Howson and colleagues showed that a high dose of trehalose administered in a short time period allows its uptake into the portal circulation, but not with low doses or a lower frequency of administration, probably because a high dose of trehalose saturates the enteric trehalase [35]. Hence, we chose a low dose to avoid direct trehalose intake, therefore favoring a peripheral action of trehalose on intestinal membranes and gut microbiota.

The loss of nigrostriatal DA neurons, which express TH, is a hallmark of neuropathological changes in the brains of PD patients [58,59,60]. Studies have also shown that the pathology in PD initiates in the terminal region of the nigrostriatal pathway, with reduced levels of TH and dopamine transporter immunostaining observed in the striatum early in the disease process [61]. Here, the overexpression of h-α-syn led to TH loss in the brain and the gut of PrP-A53T mice, with the striatum being more impacted than the SNpc. It was also previously reported that the loss of striatal TH in the PrP-A53T mice begins around 5 m of age [62]. Despite the strong overexpression of mutated α-syn A53T in DA neurons and the alteration of DA functions via the loss of TH expression, we did not observe a loss of DA neurons, such as other studies, which may be a limitation of this model [63]. This dopaminergic nigrostriatal and catecholaminergic enteric neurodegeneration was prevented by the trehalose-enriched water consumption specifically. The observed neuroprotective properties of trehalose are consistent with what was reported in PD and other models of neurodegenerative diseases [29,35,64]. However, p-S129 h α syn expression remained consistent across diets, suggesting that the neuroprotective mechanism of action of trehalose is independent of p-S129 h-α-syn modulation in this model.

PrP-A53T mice displayed motor hyperactivity, stress, anxiety, and weight loss due to the expression of A53T mutant h-α-syn. This hyperactivity, which has already been reported in the literature [34,65], may be linked to the stress and anxiety observed in PrP-A53T mice and explain their lower body weight. Moreover, Unger et al. reported a decreased expression of DA transporters in both the striatum and nucleus accumbens of PrP-A53T mice that was correlated with hyperactivity. Motor hyperactivity does not seem to result from the loss of TH but was linked to changes in DA and serotonin receptors, with the D1/D2 receptors more abundant or more active [65]. The stress, anxiety, and depressive-like phenotypes exhibited by PrP-A53T mice may be due to modulation of serotonin [66] and the overexpression of α-syn since it is known to play a role in mood regulation, and synucleinopathies are often associated with anxiety [66]. Also, one etiological basis of anxiety in PD has been linked with DA dysfunction related to dopamine transporter (DAT) expression [67,68]. Moreover, the dopaminergic system is linked to motivation and attention [69], suggesting the deficits in nesting may result from a lack of motivation. This is the first time it is reported in the PrP-A53T G2-3 mice. However, it is difficult to ascertain whether PrP-A53T mice’s anxious and stressed behavior is due to their hyperactivity or dopaminergic and serotoninergic dysfunctions. The trehalose treatment did not affect the behavioral dysfunctions observed, which suggests they are not related to TH loss, but to other pathological dysfunctions occurring in the model that were not studied here. 

The observed dysbiosis in PrP-A53T mice aligns with PD patients’ microbiota alterations, which are often correlated with disease progression, as there is a continuous decrease in fiber-degrading bacterial strains and an increase in pathobionts [8,10,11,12,15,70,71]. For example, the increased abundance of *Verrucomicrobiaceae, Akkermansiaceae* [10,70,71], *Lactobacillus* [8,10,71], and *Bifidobacterium* [8,12,71] and decreased abundance of *Faecalibacterium* [8,11,12,70] in PD patients has been demonstrated multiple times. Intestinal dysbiosis may reduce the expression of tight junction proteins, increasing intestinal permeability [72], which could explain the PrP-A53T mice constipation [55], along with the loss of enteric cholinergic neurons, since they regulate gut motility. Gastrointestinal dysfunction, such as constipation, is also observed in the early stages of PD and affects about 50% of PD patients [73]. Some gut bacteria are also able to promote α-syn aggregation and α-syn mediated pathology [74]. This evidence points to the microbiota as a major element in the pathogenesis of PD. Bacterial metabolic byproducts, including SCFAs, are often considered key candidate mediators of gut–brain communication, and altered SCFAs production has been demonstrated in a variety of neuropathologies, including PD [22,75,76]. In PrP-A53T mice, dysbiosis was associated with a decrease in the production of acetate, propionate, and isobutyrate. This is in accordance with several studies that report a decrease in the concentration of SCFAs in patients and models of PD [11,12,74]. This loss in SCFAs could also be involved in PrP-A53T mice constipation since SCFAs can affect gut peristalsis [77,78].

Trehalose treatment modulated the gut microbiota composition and diversity. Of interest, *Lachnospiraceae* and *Ruminococcaceae* families, SCFA producers and members of the *Firmicutes* phylum, were strongly enriched by trehalose consumption in PrP-A53T mice. Those two populations, potentially beneficial for intestinal permeability and cognitive symptoms, are less present in PD patients [70,71,79,80] and in mice receiving fecal transplants from PD patients [74]. This loss is thought to increase intestinal permeability, decrease production of SCFAs, and is correlated with motor and cognitive symptoms [70,80]. So, the PrP-A53T genotype had an impact on the structure and composition of the mice’s gut microbiota, and the consumption of trehalose favors the abundance of the genus *Lachnospiraceae* and *Ruminococcaceae*, producers of mainly butyrate and acetate [81]. Paradoxically, the increase observed in the present study was not associated with the expected rise in SCFAs in PrP-A53T mice feces at 6 or 9.5 m.

We observed lower levels of the incretin glucagon-like peptide 1 (GLP-1) along with a lower glycemia in PrP-A53T mice compared to NC mice, which could be linked to a loss of dopaminergic neurons, which in turn is known to alter glycemic control [82]. Moreover, α-syn plays a role in glucose regulation, function of beta cells, and glucose utilization in peripheral tissues, suggesting an association between synucleinopathies and glucose dysregulation [82]. Several studies also suggest a link between diabetes mellitus (DM) and PD: patients with DM have a 40% higher risk of developing PD than normoglycemic subjects. A potential reason for this association is that within the CNS, insulin is able to modulate many processes disrupted in PD, including autophagy, oxidative stress, and α-syn aggregation [83,84], mainly via the activation of the MAPK/ERK and PI3K/AKT pathways. It has been shown that insulin affects TH and DAT expression, which in turn influences DA levels [83,85]. Also, a shared mechanism between PD and type 2 diabetes mellitus (T2DM) is protein aggregation, with α-syn aggregation in PD and amylin aggregation in insulin-producing cells for diabetes [84,86].

In the present study, trehalose shows a hypoglycemic effect in NC mice. It is coherent with previous studies reporting that trehalose prevents a spike in blood glucose levels and induces lower insulin secretion compared to glucose administration [87,88]. Moreover, in humans, oral trehalose intake at 3.3 g/day for 3 m reduces inflammation and improves the quality of life in patients with diabetes. It also improves glucose tolerance in patients with high body mass index, modulating sugar absorption so as to avoid glucose peaks [89]. Trehalose may inhibit multiple intestinal glucose transporters (GLUT/SLC2A), slowing glucose entry into the small intestine and leading to a reduction in glucose absorption [88,90]. Coupled with the observed induction of GLP-1 secretion by trehalose, this may increase insulin and lower blood glucose levels. GLP-1 is a gut-derived peptide secreted in response to ingested nutrients, specifically fibers, peptides, amino acids, and unsaturated fatty acids [91] in the distal small intestine and colon [92]. One of its many functions is to increase glucose metabolism after food intake, regulating glucose-dependent insulin secretion by pancreatic beta cells [93]. GLP-1 can slow gastric emptying, which may also reduce glycemic variations [94]. GLP-1 can also exert inhibitory effects on gastrointestinal tract (GIT) motility, directly influencing the GIT wall and improving myenteric neuron survival [54,95]. Furthermore, GLP-1 and its analogs have been shown to modulate DA neurotransmission in different regions of the CNS [96,97]. GLP-1 has also been proposed as a neuroprotective hormone [98], and GLP-1 analogs have demonstrated protective effects in humans [99,100,101,102]. GLP-1 binds to GLP-1 receptors (GLP-1R), a G-protein-coupled transmembrane receptor, expressed in the periphery as well as most regions of the brain, primarily in neurons [99]. However, endogenous GLP-1 is rapidly degraded and inactivated (plasma half-life of 1–2 min) by the circulating enzyme dipeptidyl peptidase IV (DPP-IV), resulting in the formation of a metabolite that does not activate GLP-1R [103]. As a result, the concentration of intact endogenous GLP-1 in the circulation is very low and perhaps not sufficient to directly activate central GLP-1Rs [104]. Thus, central GLP-1Rs are probably targets for GLP-1 produced by brain stem neurons in the nucleus of the solitary tract. It would be interesting to see if the secretion of central GLP-1 is also impacted in the PrP-A53T mouse model and restored by trehalose. This widespread cerebral distribution of GLP-1Rs suggests that GLP-1 might be involved in the regulation of multiple neurologic and cognitive functions that extend beyond the regulation of glucose metabolism. In this regard, GLP-1R agonists and GLP-1 analogs (exenatide, liraglutide, and lixisenatide), used to treat diabetes, have been identified as promising drugs for PD, showing neuroprotective and neurorestorative properties. For example, exenatide intake [102] reduced the severity of symptoms in PD patients; NLY01 had a beneficial effect on young patients with PD (aged less than 60 years) [86]; lixisenatide and liraglutide showed neuroprotective effects in the MPTP mouse model of PD by preventing motor impairment and reduction in TH levels in the SNpc and basal ganglia [101]. They act on inflammatory processes and microglia, could stimulate neurogenesis [86,105], afford protection against dopaminergic neurodegeneration, and improve motor function in various models of PD [99,100,101,102]. GLP-1R agonists have also been shown to improve intestinal epithelial integrity and reduce increased gut permeability [106]. Therefore, GLP-1 could be involved in the mechanism of action of trehalose, and we showed that trehalose consumption prevented the loss of GLP-1 secretion. Besides, this increase in the secretion of GLP-1 by trehalose is also observed in humans [88].

Recent studies have shown that it is possible that orally administered trehalose is not entirely broken down into glucose, and a small quantity of non-metabolized trehalose could find its way into the bloodstream after ingestion and have an independent physiological action, mainly when large quantities are ingested [35,57,88,107]. Despite the low dose chosen in this study, we cannot rule out that not all administered trehalose was catabolized in the gut and that some trehalose may have crossed into the peripheral circulation to have direct neuroprotective effects in the periphery as a disaccharide, highlighting an avenue for future investigation.

Trehalose has been authorized in the food industry in the United States since 2000, in the European market since 2001, and in Canada since 2005 [108]. According to the toxicological results available in humans, the tolerance is good during food consumption [109]. Over the last decades, the cost of industrial synthesis of trehalose has reduced; consequently, there was an increase in the consumption of trehalose as a dietary sugar [107]. Concurrently, *Clostridioides difficile* (CD) ribotypes 027 and 078 emerged as causes of epidemics. A link was proposed between this emergence and commercialization of trehalose since CD 027 and 078 evolved the capacity to catabolize trehalose to fuel growth in the absence of glucose [110]; however, current epidemiologic and clinical data do not support this hypothesis [108,111]. Therefore, there are no contraindications to the use of trehalose, which has shown a promising nutraceutical potential. Trehalose consumption, combined with other treatments, could target the early stages of PD and slow its progression.

## 5. Conclusions

To conclude, this study provides evidence that dietary intake of trehalose can offer central and enteric neuroprotection against synucleinopathy-induced neurodegeneration, using the progressive transgenic mouse model PrP-A53T. We are the first to propose that neuroprotection with trehalose could be partially mediated by its ability to enhance GLP-1 secretion in the ENS. GLP-1′s multifaceted functions in glucose metabolism, insulin regulation, gut motility, and neuroprotective properties emphasize its potential role in trehalose’s mechanism. The effect of trehalose could also be mediated by the modulation of the gut microbiota, particularly increasing the *Lachnospiraceae* population, but also influencing microbial diversity. A longitudinal study of the microbiota over the time would be interesting to confirm the impact of trehalose on the microbiota. While our study provides valuable insights into the neuroprotective effects of trehalose, it is essential to acknowledge certain limitations. The possibility of orally administered trehalose not being entirely metabolized in the gut raises questions about potential direct neuroprotective effects in the periphery as a disaccharide. Further investigations into the intricate interplay of trehalose with the gut microbiota, GLP-1, and glycemic control hold the potential for advancing therapeutic strategies in PD and related neurodegenerative disorders. We cannot exclude that trehalose could also promote alternative mechanisms for neuronal protection. Therefore, further studies are required for a more comprehensive investigation of the underlying mechanisms of trehalose, such as a direct neuroprotective effect of trehalose as a disaccharide, for example, by inhibiting trehalase.

## Figures and Tables

**Figure 1 nutrients-16-03309-f001:**
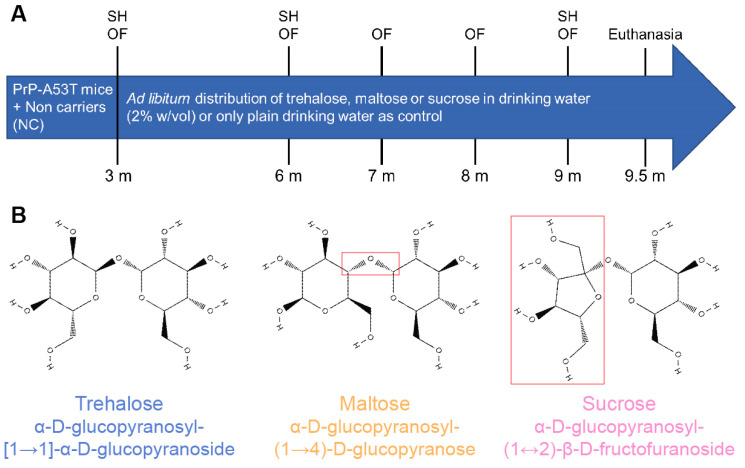
(**A**) Experimental timeline in mice showing the distribution of plain water or water mixed with 2% *w*/*v* of trehalose, maltose, or sucrose and in vivo tests performed. Stool harvest (SH); open field (OF); month (m). (**B**) Chemical formula of administered sugars: trehalose, maltose, and sucrose. Differences between the formula of trehalose and other sugars are framed in red.

**Figure 2 nutrients-16-03309-f002:**
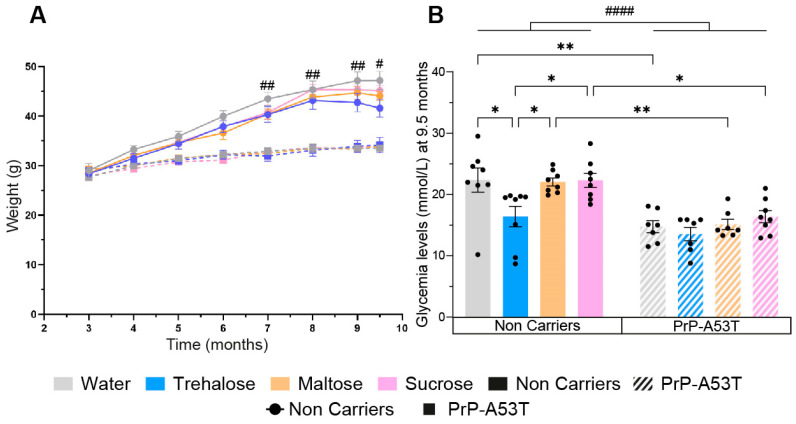
Effect of genotype and treatments on mice weight and glycemia. (**A**) Time course of mice’s weight from 3 m to 9.5 m of age. (**B**) Effect of sugar administration on blood sugar levels at the moment of euthanasia when mice are 9.5 m old. n = 7–8 mice per group. # *p* < 0.05, ## *p* < 0.01, #### *p* < 0.0001 for the genotype effect (NC vs. PrP-A53T), * *p* < 0.05, ** *p* < 0.01; two-way ANOVA, Tukey test.

**Figure 3 nutrients-16-03309-f003:**
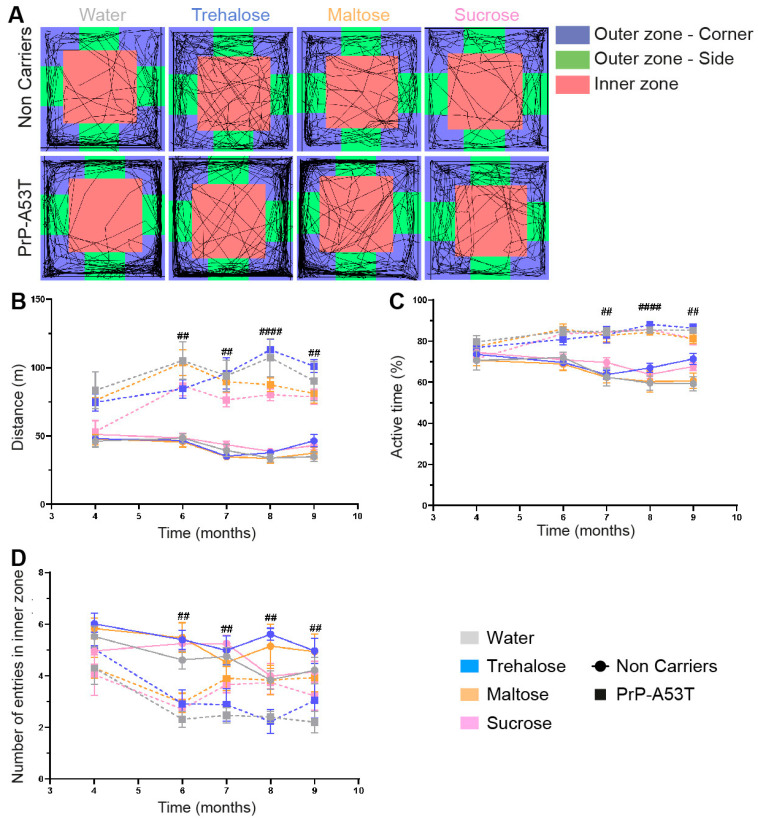
Effect of genotype and treatments on motor behavior. (**A**) Representation of mice trajectories during the open field test (30 min) at 9.5 m of age. (**B**) Analysis of the total distance traveled by mice in the open field test from 3 m to 9.5 m of age. (**C**) Analysis of the activity levels of mice during the open field test by measuring the percentage of active time from 3 m to 9.5 m of age. (**D**) Number of entries in the open field inner zone to measure anxiety-like behavior from 3 m to 9.5 m of age. n = 7–8 mice per group. ## *p* < 0.01, #### *p* < 0.0001 for the genotype effect (NC vs. PrP-A53T); two-way ANOVA, Tukey test.

**Figure 4 nutrients-16-03309-f004:**
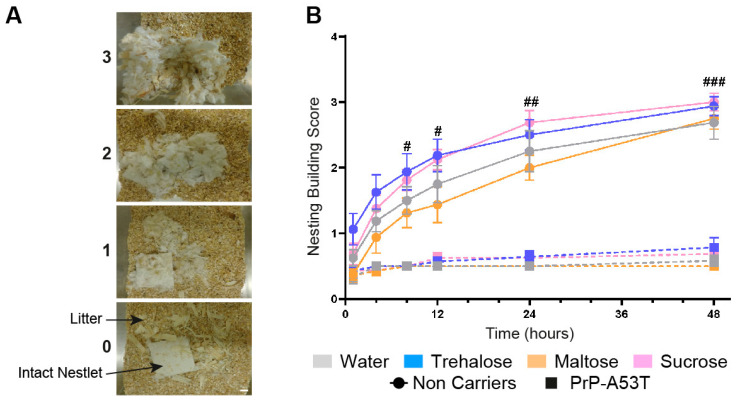
Effect of genotype and treatments on anxiety and depression. (**A**) Representative images of nest building scores. Scale bar = 1 cm. (**B**) Nest-building scores were assessed at 1, 4, 8, 12, 24, and 48 h. n = 7–8 mice per group. # *p* < 0.05, ## *p* < 0.01, ### *p* < 0.001 for the genotype effect (NC vs. PrP-A53T); two-way ANOVA, Tukey test.

**Figure 5 nutrients-16-03309-f005:**
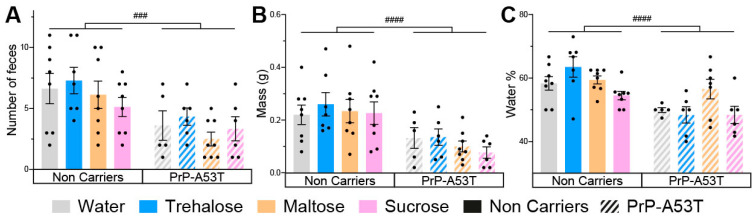
Effect of genotype and treatments on mice stool. (**A**) Fecal boli count of mice in a 30-min time period at 9.5 m of age. (**B**) Total wet mass of the feces harvested at 9.5 m of age. (**C**) Percentage of water present in the harvested feces at 9.5 m of age. n = 7–8 mice per group. ### *p* < 0.001, #### *p* < 0.0001 for the genotype effect (NC vs. PrP-A53T); two-way ANOVA, Tukey test.

**Figure 6 nutrients-16-03309-f006:**
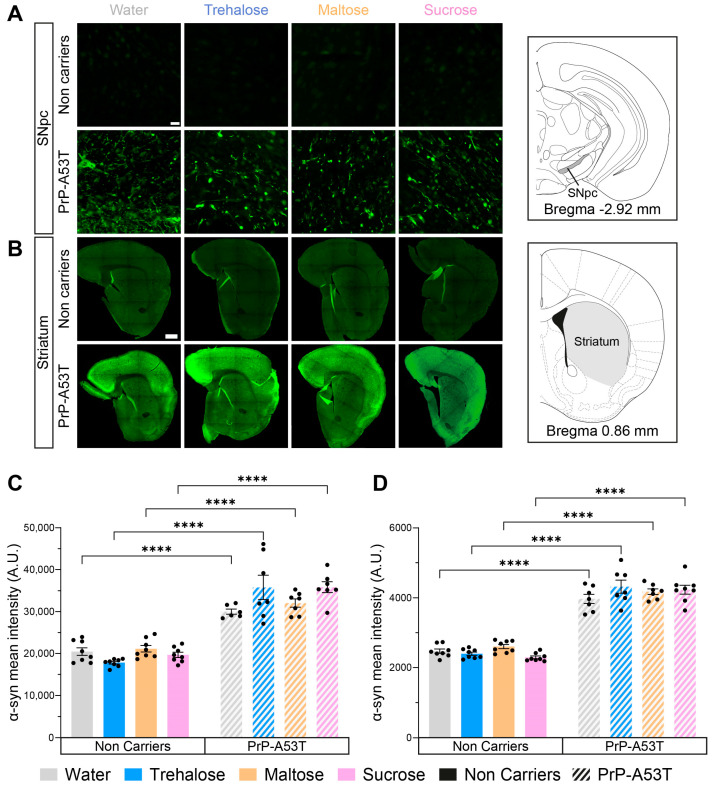
Effect of genotype and treatments on phosphorylated S129 human α-syn mean intensity in the substantia nigra pars compacta (SNpc) and the striatum. (**A**) Representative example of α-syn immunofluorescence in the SNpc. Scale bar = 20 µm. (**B**) Representative example of striatal α-syn immunofluorescence. Scale bar = 500 µm. (**C**) Phosphorylated S129 human α-syn mean intensity in the SNpc in PrP-A53T mice and NC. Values shown are the mean pixel intensity ± SEM of n = 7–8 mice per group. (**D**) Striatal phosphorylated S129 human α-syn mean intensity in PrP-A53T mice and NC. Values shown are the mean pixel intensity ± SEM of 7–8 mice per group. **** *p* < 0.0001; two-way ANOVA, Tukey test.

**Figure 7 nutrients-16-03309-f007:**
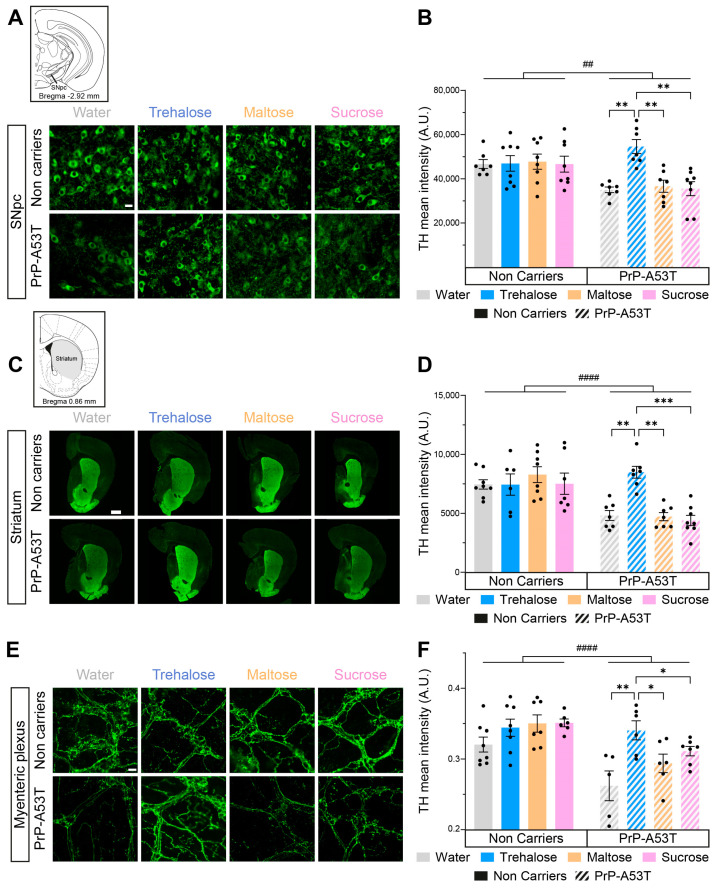
Effect of genotype and treatments on TH in the substantia nigra pars compacta (SNpc), the striatum and the myenteric plexus. (**A**) Representative microphotographs of TH immunofluorescence in the SNpc. Scale bar = 20 µm. (**B**) TH mean intensity in the SNpc in PrP-A53T mice and NC. Values shown are the mean pixel intensity ± SEM of n = 7–8 mice per group. (**C**) Representative microphotographs of striatal TH immunofluorescence. Scale bar = 500 µm. (**D**) Striatal TH mean intensity in PrP-A53T mice and NC. Values shown are the mean pixel intensity ± SEM of 7–8 mice per group. (**E**) Representative microphotographs of TH immunofluorescence in the myenteric plexus. Scale bar = 20 µm. (**F**) TH mean intensity in the myenteric plexus in PrP-A53T mice and NCs. Values shown are the mean pixel intensity ± SEM of 7–8 mice per group. * *p* < 0.05, ** *p* < 0.01, *** *p* < 0.001, ## *p* < 0.01, #### *p* < 0.0001 for the genotype effect (NC vs. PrP-A53T); two-way ANOVA, Tukey test.

**Figure 8 nutrients-16-03309-f008:**
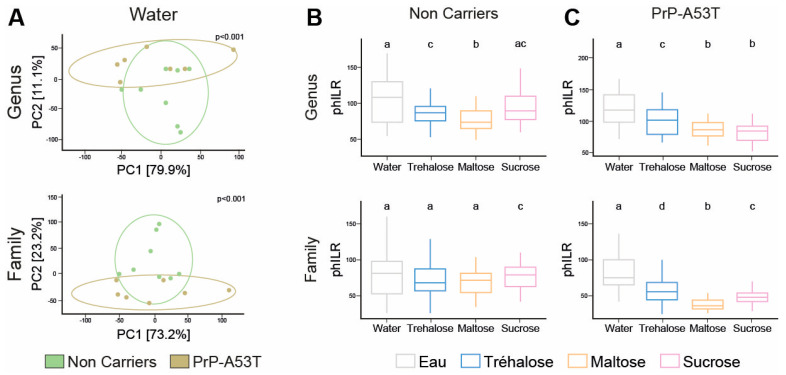
Variation in gut microbiota composition by measuring the phylogenetic beta diversity between the various groups of mice. (**A**) PCA of Euclidian distances calculated using PhILR-transformed abundances for the NCs and PrP-A53T mice treated with water at the genus and family levels. Each point represents a unique gut microbiome sample (n = 7–8). Ellipses show the normal-theory confidence regions. Principal components one and two explained 79.9% and 11.1% of the variation in gut microbiome structure at the genus level and 73.2% and 23.2% at the family level, respectively. *p* < 0.001 for each graph. (**B**) Beta-diversity boxplots and indices for water, trehalose, maltose, and sucrose in NC-treated mice at the genus and family levels. Tukey HSD test group letters (*p* < 0.05) are represented: groups with the same letter are not different; groups with different letters are significantly different. (**C**) Beta-diversity boxplots and indices for water, trehalose, maltose, and sucrose in PrP-A53T-treated mice at the genus and family levels. n = 7–8 mice per group. Tukey HSD test group letters (*p* < 0.05) are represented: groups with the same letter are not different; groups with different letters are significantly different.

**Figure 9 nutrients-16-03309-f009:**
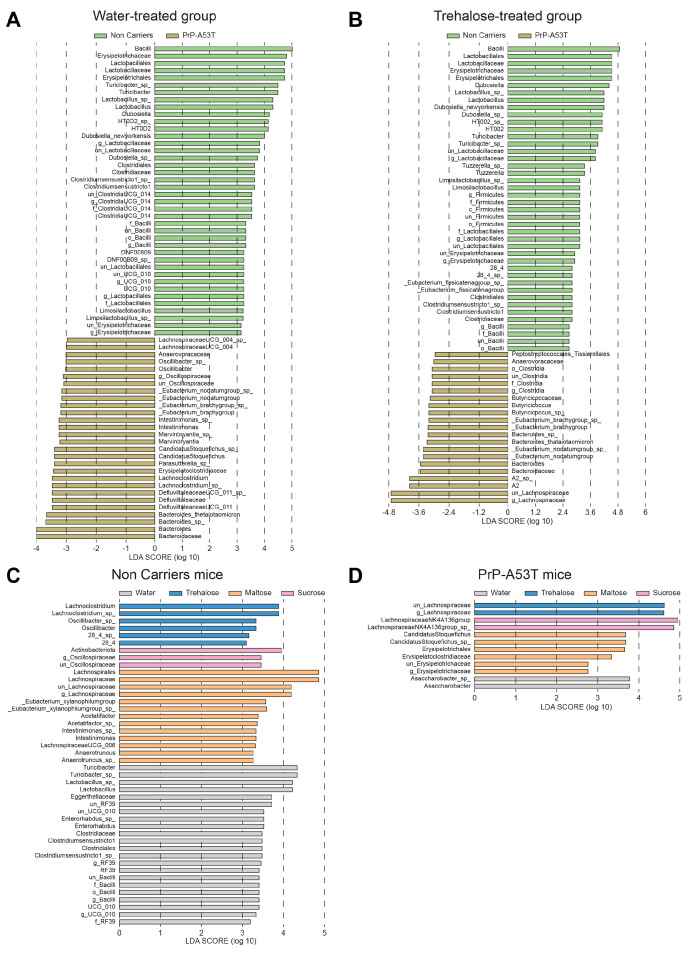
Effect of genotype and treatment on the relative abundance of taxa in NC and PrP-A53T mice using LEfSe analysis. (**A**) Relative abundance of bacteria with LDA scores of differentially abundant operational taxonomic units (OTUs) among NC mice (green) or PrP-A53T mice (brown) treated with normal water. (**B**) Relative abundance of bacteria with LDA scores of differentially abundant OTUs among NC mice (green) or PrP-A53T mice (brown) treated with trehalose. (**C**) Relative abundance of bacteria with LDA scores of differentially abundant OTUs among NC mice treated with normal water (grey), trehalose (blue), maltose (orange), and sucrose (rose). (**D**) Relative abundance of bacteria with LDA scores of differentially abundant OTUs among PrP-A53T mice treated with normal water (grey), trehalose (blue), maltose (orange), and sucrose (rose). The LDA scores represent the effect size of each abundant OTUs. Species enriched in each group with an LDA score > 2 are considered. n = 7–8 mice per group.

**Figure 10 nutrients-16-03309-f010:**
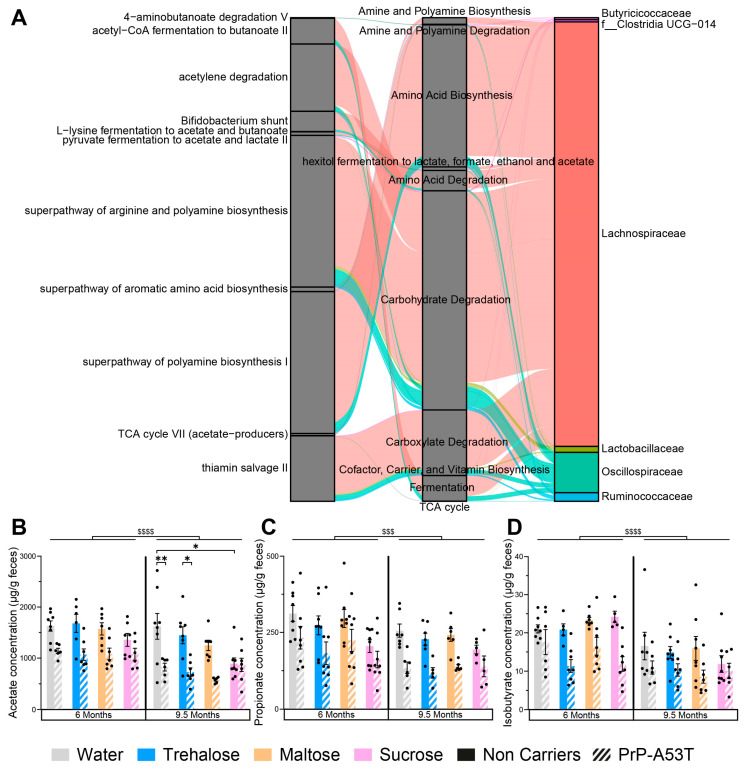
Effect of genotype and treatments on gut microbiota functional markers and metabolic pathways. (**A**) Sankey diagram (or alluvial plot) showing the bacterial contribution to predicted functional markers in the PrP-A53T trehalose-treated mice, from feces harvested at 9.5 m of age. Left: functional microbiota selected markers. Middle: functional class or superclass. Right: contribution of the most abundant families in the microbiota. The wider the band, the greater the contribution. The functional fraction was calculated by accumulating the genome coverage values of genomes from a specific microbial group that possesses a given functional trait. The width of curved lines from a specific microbial group to a given functional trait indicates their corresponding proportional contribution to a specific metabolism. (**B**) Fecal acetate contents at 6 m and 9.5 m of age. (**C**) Fecal propionate contents at 6 m and 9.5 m of age. (**D**) Fecal isobutyrate contents at 6 m and 9.5 m of age. n = 7–8 mice per group. * *p* < 0.05, ** *p* < 0.01, $$$ *p* < 0.001, $$$$ *p* < 0.0001 for the age effect (6 m vs. 9.5 m); three-way ANOVA, Tukey test.

**Figure 11 nutrients-16-03309-f011:**
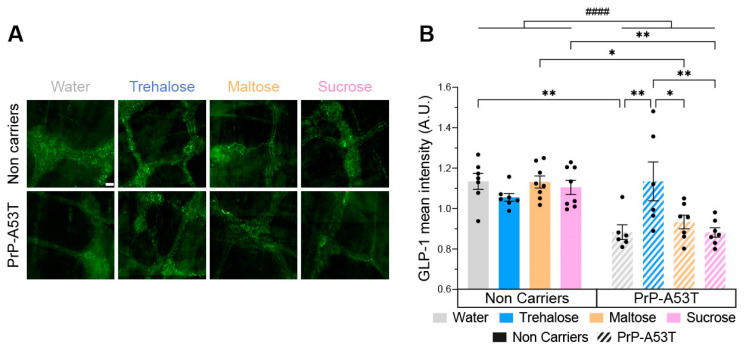
Effect of genotype and treatments on glucagon-like peptide 1 (GLP-1) immunoreactivity in the myenteric plexus. (**A**) Representative example of GLP-1 immunofluorescence in the myenteric plexus. Scale bar = 20 µm. (**B**) GLP-1 mean intensity in the myenteric plexus in PrP-A53T mice and NCs. Values shown are the mean pixel intensity ± SEM of n = 7–8 mice per group. * *p* < 0.05, ** *p* < 0.01, #### *p* < 0.0001 for the genotype effect (NCs vs. PrP-A53T); two-way ANOVA, Tukey test.

**Figure 12 nutrients-16-03309-f012:**
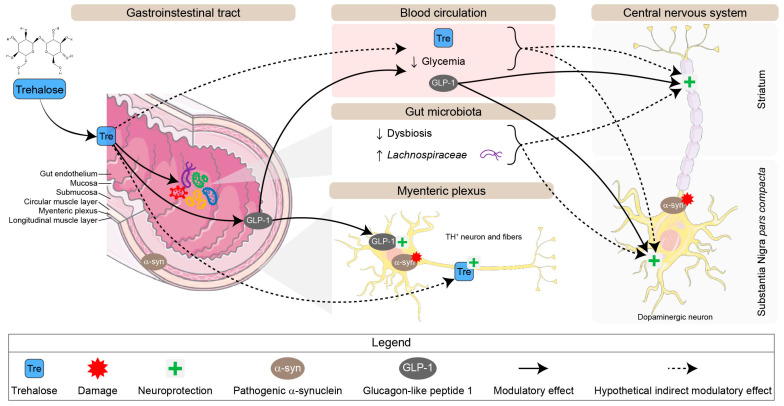
Schematic representation of potential neuroprotective pathways for trehalose in the enteric and central nervous systems in the PrP-A53T mouse model. Abbreviation: TH, tyrosine hydroxylase.

**Table 1 nutrients-16-03309-t001:** Oligonucleotide Sequences for Amplification.

Primer	Sequence
V3-V4_F_341F (first PCR)	ACACTCTTTCCCTACACGACGCTCTTCCGATCTCCTACGGGNGGCWGCAG
V3-V4_R_805R (first PCR)	GTGACTGGAGTTCAGACGTGTGCTCTTCCGATCTGACTACHVGGGTATCTAATC
Generic forward second-PCR primer	AATGATACGGCGACCACCGAGATCTACAC[index1]ACACTCTTTCCCTACACGAC
Generic reverse second-PCR primer	CAAGCAGAAGACGGCATACGAGAT[index2]GTGACTGGAGTTCAGACGTGT

**Table 2 nutrients-16-03309-t002:** Primary and Secondary Antibodies.

**Primary antibodies**	**Host**	**Company**	**Catalog #**	**Dilution**
S129 phosphorylated α-synuclein	Rabbit	Abcam	Ab51253	1/200
Tyrosine Hydroxylase	Sheep	Invitrogen	PA1-4679	1/1000
Glucagon like peptide-1	Rabbit	Invitrogen	PA5-79303	1/750
Anti-Choline Acetyltransferase	Goat	Sigma-Aldrich	AB144P	1/100
**Secondary antibodies**	**Host**	**Company**	**Catalog #**	**Dilution**
Anti-Rabbit IgG Alexa Fluor^TM^ 647	Donkey	Life technologies	A31573	1/1000
Anti-Sheep IgG Alexa Fluor^TM^ 647	Donkey	Invitrogen	A21448	1/1000
Anti-Rabbit IgG Alexa Fluor^TM^ 488	Donkey	Life Technologies	A21206	1/1000
Anti-Goat IgG Alexa FluorTM 633	Donkey	Life Technologies	A21082	1/1000

## Data Availability

Data generated or analyzed during this study are included in this published article and its Appendix A. Data will be made available on reasonable request.

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
