# Peer review of "Oral Trehalose Intake Modulates the Microbiota–Gut–Brain Axis and Is Neuroprotective in a Synucleinopathy Mouse Model"

_nutrients, 2024, doi:10.3390/nu16193309_

Round 1
Reviewer 1 Report
Comments and Suggestions for Authors
The authors have studied the neuroprotective effects of trehalose in the PrP-A53T transgenic mice that constitute a model of synucleinopathy. The starting hypothesis is that trehalose acts on the microbiota-gut-brain axis. Exposure to sugar would modulate the composition and activity of the intestinal microbiota that would act through the gut-brain axis to produce neuroprotective effects.
The study is interesting and well thought out. However, there are few results indicating a direct neuroprotective effect of trehalose, as no effects are observed in the in vivo animal behavioral tests directly related to the symptoms of Parkinson's disease.
The article is eligible for publication, but the reviewer makes a few comments and suggestions that can be considered to enrich the manuscript.
RESULTS
1. The authors report that trehalose has a hypoglycemic effect, but only in NC mice. How can it be interpreted? Why is glycemia lower in PrP-A53T than in NC mice? Could you please discuss this point in more depth?
2. The authors see no effects of Tre on the different motor and non-motor symptoms (anxiety and depressive behaviors). However, there are consistent effects of Tre in PrP-A53T mice on TH expression in the myenteric plexus, substantia nigra, and striatum. In this sense, Tre prevents neurodegeneration. Please discuss this topic a little more.
3. GLP-1 levels in the myenteric plexus are lower in PrP-A53T mice than in NC and Tre has a positive effect only in transgenic mice. Do you have any hypothesis as to the reason for these differences between the two strains?
DISCUSSION
4. Figure 12 is very clear and shows a good summary of the possible neuroprotective pathways of trehalose in the enteric and central nervous systems in PrP-A53T mice. The authors comment that part of Tre can reach the myenteric plexus or the blood circulation unchanged. How does this process take place? On the other hand, it is surprising that Tre has an effect, but mannose does not, considering that the two disaccharides are made up of two glucose molecules. Please discuss this difference a little more.
5. The study shows that Tre intake does not affect the behavior of mice (motor and non-motor) (lines 615-617). Based on these results, can it be stated that the intake of this sugar can improve the symptoms of Parkinson's disease?
6. In lines 649-650, the authors state that the various sugars tested have no effect on SCFA synthesis. Later (lines 653-657) they say it may be because the effect on SCFAs occurs before the time they get the samples. It's a somewhat speculative statement. Please discuss a little more the consequences of the observed changes in the microbiota.
Minor comments:
7. Line 153. Please, check the units in which the fecal samples collected are expressed.
8. Line 199. Please, check the section title.
9. Lines 289-290. This sentence is difficult to understand. Rewriting is recommended.
10. Legend of Figure 3. Lines 334-336. Please check the letters C and D. They seem to have been interchanged. The “E” in the legend is not clear, nor is it in line 342.
11. Legend of Figure 9. Authors refer to identified species. Wouldn't it be more appropriate to refer to OTUs?
12. Lines 642-648. Please, simplify this part as there are repeated concepts.
Author Response
Manuscript ID nutrients-3145945
Type Article
Title Oral Trehalose Intake modulates the Microbiota-Gut-Brain Axis and is neuroprotective in a Synucleinopathy Mouse Model
Modifications in the manuscript have been highlighted in yellow.
REVIEWER #1
General comment: The authors have studied the neuroprotective effects of trehalose in the PrP-A53T transgenic mice that constitute a model of synucleinopathy. The starting hypothesis is that trehalose acts on the microbiota-gut-brain axis. Exposure to sugar would modulate the composition and activity of the intestinal microbiota that would act through the gut-brain axis to produce neuroprotective effects.
The study is interesting and well thought out. However, there are few results indicating a direct neuroprotective effect of trehalose, as no effects are observed in the in vivo animal behavioral tests directly related to the symptoms of Parkinson's disease.
The article is eligible for publication, but the reviewer makes a few comments and suggestions that can be considered to enrich the manuscript.
Reply: We thank the reviewer for the general constructive comments.
RESULTS
Comment 1. The authors report that trehalose has a hypoglycemic effect, but only in NC mice. How can it be interpreted? Why is glycemia lower in PrP-A53T than in NC mice? Could you please discuss this point in more depth?
Reply 1. Glycemia is lower in PrP-A53T than in NC mice. We don’t have a robust explanation about it. May be the PrP-A53T mice being more overactive, they use more glucose than NC mice, leading to a lower glycemia. Since this motor hyperactivity is paradoxical, supposedly a motor activity artifact from the Prp-A53T related to dopamine signaling compensation, we cannot speculate about the origin of this hypoglycemia. However, hypoglycemia has been reported in synucleinopathy models (Chmiela et al., 2022), and it has been hypothesize that alpha-synuclein can interfere with glucose regulation and utilization in peripheral tissues (see the discussion in the manuscript).
Comment 2. The authors see no effects of Tre on the different motor and non-motor symptoms (anxiety and depressive behaviors). However, there are consistent effects of Tre in PrP-A53T mice on TH expression in the myenteric plexus, substantia nigra, and striatum. In this sense, Tre prevents neurodegeneration. Please discuss this topic a little more.
Reply 2. Indeed, in accordance with the literature, it was expected that trehalose would show neuroprotective effects on TH expression. However, the proposed mechanisms are frequently related to autophagy processes in vitro. There are few reports about the neuroprotective effects of trehalose in Parkinson’s disease models in vivo. For instance, in rats, an increase in trehalose-mediated autophagy is reported (He et al., 2016). Of great interest, in C57/bl6 mice treated with MPTP, trehalose reduced the loss of TH and DAT expression in caudate putamen and substantia nigra (Sarkar et al., 2014). Moreover, in a rotenone model in rats, trehalose treatments resulted in the reduction of dopaminergic neuronal degeneration in the substantia nigra (Wu et al., 2015). See the discussion in the manuscript.
Comment 3. GLP-1 levels in the myenteric plexus are lower in PrP-A53T mice than in NC and Tre has a positive effect only in transgenic mice. Do you have any hypothesis as to the reason for these differences between the two strains?
Reply 3. This is an interesting comment. However, we cannot explain why GLP-1 expression is lower in Prp-A53T mice vs NC mice. We can speculate that there is a link between the observed lower expression of GLP-1 and motor hyperactivity in Prp-A53T mice compared to NC mice. Transgenic mice may require more glucose to sustain their motor hyperactivity, therefore less GLP-1 would be needed? At the moment, there is no available explanation in the literature about this phenomenon. We decided to not discuss it further in the manuscript since it might be too speculative.
DISCUSSION
Comment 4. Figure 12 is very clear and shows a good summary of the possible neuroprotective pathways of trehalose in the enteric and central nervous systems in PrP-A53T mice. The authors comment that part of Tre can reach the myenteric plexus or the blood circulation unchanged. How does this process take place? On the other hand, it is surprising that Tre has an effect, but mannose [this reviewer meant maltose?] does not, considering that the two disaccharides are made up of two glucose molecules. Please discuss this difference a little more.
Reply 4. Very few intact trehalose could enter the blood stream following an oral administration considering that there are no trehalose transporters expressed in mammals and that most trehalose is metabolized in 2 molecules of glucose in the gut. Indeed, we cannot rule out that if the amount of ingested trehalose is high enough, around 1% could enter passively the blood stream, as represented in Figure 12. As highlighted by this reviewer, maltose treatment has no neuroprotective effect in the present study, suggesting that glucose originating by either trehalose or maltose are not the neuroprotective agents. Therefore, trehalose effects are mediated by another mechanism, specific to trehalose. As discussed in the manuscript, trehalose may promote GLP-1 secretion and change the intestinal microbiota profile and related metabolome, leading to potential neuroprotective effects.
Comment 5. The study shows that Tre intake does not affect the behavior of mice (motor and non-motor) (lines 615-617). Based on these results, can it be stated that the intake of this sugar can improve the symptoms of Parkinson's disease?
Reply 5. The intake of trehalose prevents the loss of TH expression in the present study. Since there is an artifactual motor hyperactivity in the Prp-A53T, this should not be taken into account in the interpretation of the results. As a reminder, Prp-A53T mice show motor hyperactivity related to a decreased expression of dopamine transporters in the striatum and nucleus accumbens (Unger et al., 2006), as a compensatory mechanism. This motor hyperactivity is highly reduced the week prior to euthanasia, since the animals are getting very sick and reach their ethical end points. Therefore, it is not possible to assess whether the motor hyperactivity would still be observed later despite the ongoing neurodegenerative processes (loss of TH expression). We cannot translate directly our result to patients’ symptoms with this Prp-A53T mouse model.
Comment 6. In lines 649-650, the authors state that the various sugars tested have no effect on SCFA synthesis. Later (lines 653-657) they say it may be because the effect on SCFAs occurs before the time they get the samples. It's a somewhat speculative statement. Please discuss a little more the consequences of the observed changes in the microbiota.
Reply 6. The comment in lines 653-657 is based on a previous published study (Miura et al., 2021) in which it has been reported that trehalose-mediated gut SCFA alterations occurred only very early (22 days after the beginning of the treatment with trehalose) and were not seen later any longer. As mentioned by the reviewer, this section might look somewhat speculative, so we decided to remove this statement since we don’t see any major changes in SCFA contents at both 3 and 6 months from the beginning of trehalose treatments.
Minor comments:
Comment 7. Line 153. Please, check the units in which the fecal samples collected are expressed.
Reply 7. Done.
Comment 8. Line 199. Please, check the section title.
Reply 8. Done.
Comment 9. Lines 289-290. This sentence is difficult to understand. Rewriting is recommended.
Reply 9. Done.
Comment 10. Legend of Figure 3. Lines 334-336. Please check the letters C and D. They seem to have been interchanged. The “E” in the legend is not clear, nor is it in line 342.
Reply 10. Done.
Comment 11. Legend of Figure 9. Authors refer to identified species. Wouldn't it be more appropriate to refer to OTUs?
Reply 11. Done.
Comment 12. Lines 642-648. Please, simplify this part as there are repeated concepts.
Reply 12. Done.
References
Chmiela, T., WÄ™grzynek, J., Kasprzyk, A., Waksmundzki, D., Wilczek, D., Gorzkowska, A., 2022. If Not Insulin Resistance so What? - Comparison of Fasting Glycemia in Idiopathic Parkinson’s Disease and Atypical Parkinsonism. Diabetes Metab Syndr Obes 15, 1451–1460. https://doi.org/10.2147/DMSO.S359856
He, Q., Koprich, J.B., Wang, Y., Yu, W., Xiao, B., Brotchie, J.M., Wang, J., 2016. Treatment with Trehalose Prevents Behavioral and Neurochemical Deficits Produced in an AAV α-Synuclein Rat Model of Parkinson’s Disease. Mol Neurobiol 53, 2258–2268. https://doi.org/10.1007/s12035-015-9173-7
Miura, H., Mukai, K., Sudo, K., Haga, S., Suzuki, Y., Kobayashi, Y., Koike, S., 2021. Effect of trehalose supplementation in milk replacer on the incidence of diarrhea and fecal microbiota in preweaned calves. J Anim Sci 99, skab012. https://doi.org/10.1093/jas/skab012
Sarkar, S., Chigurupati, S., Raymick, J., Mann, D., Bowyer, J.F., Schmitt, T., Beger, R.D., Hanig, J.P., Schmued, L.C., Paule, M.G., 2014. Neuroprotective effect of the chemical chaperone, trehalose in a chronic MPTP-induced Parkinson’s disease mouse model. Neurotoxicology 44, 250–262. https://doi.org/10.1016/j.neuro.2014.07.006
Unger, E.L., Eve, D.J., Perez, X.A., Reichenbach, D.K., Xu, Y., Lee, M.K., Andrews, A.M., 2006. Locomotor hyperactivity and alterations in dopamine neurotransmission are associated with overexpression of A53T mutant human alpha-synuclein in mice. Neurobiol Dis 21, 431–443. https://doi.org/10.1016/j.nbd.2005.08.005
Wu, F., Xu, H.-D., Guan, J.-J., Hou, Y.-S., Gu, J.-H., Zhen, X.-C., Qin, Z.-H., 2015. Rotenone impairs autophagic flux and lysosomal functions in Parkinson’s disease. Neuroscience 284, 900–911. https://doi.org/10.1016/j.neuroscience.2014.11.004

Reviewer 2 Report
Comments and Suggestions for Authors
The current study comprises of vast number of parameters estimated but I have one major concern related to behavioral study. As mentioned in OFT, was the timeline based on animal age or the starting point of the experiment? Isn’t 30 minutes too long for OFT for each animal? Why didn’t the authors perform other behavioral tests on specific motor functions, for example, rotarod, beam balance, etc.? Furthermore, OFT and nesting tests evaluate anxiety and depression rather than motor dysfunction associated with Parkinson’s disease. What was the point and rationale of performing these tests instead?
A few other comments are as follows.
§ Why is Parkinson’s disease not mentioned in the title? I am NOT asking to change it, but want clarification because synucleopathy is the main feature of PD.
§ Did you perform a power analysis before deciding on the sample size?
§ Were the animals kept individually or in groups in cages? How the water consumption by each animal was estimated if kept together? The dose should be justified and almost equal for all the animals.
§ May I know, what was the euthanasia method?
Comments on the Quality of English LanguageMinor editing is required.
Author Response
Manuscript ID nutrients-3145945
Type Article
Title Oral Trehalose Intake modulates the Microbiota-Gut-Brain Axis and is neuroprotective in a Synucleinopathy Mouse Model
Modifications in the manuscript have been highlighted in yellow.
REVIEWER #2
General comments: The current study comprises of vast number of parameters estimated but I have one major concern related to behavioral study. As mentioned in OFT, was the timeline based on animal age or the starting point of the experiment? Isn’t 30 minutes too long for OFT for each animal? Why didn’t the authors perform other behavioral tests on specific motor functions, for example, rotarod, beam balance, etc.? Furthermore, OFT and nesting tests evaluate anxiety and depression rather than motor dysfunction associated with Parkinson’s disease. What was the point and rationale of performing these tests instead?
Reply to general comments: We thank the reviewer for his positive comments. Regarding OFT, the timeline refers to the age of the mice, as indicated in the legends. OFTs were performed for 30 minutes, as regularly done in our lab and elsewhere. Of importance, OFT were also analyzed for the 10 first minutes and the 10 last minutes of the video recording, and there was no early hyperactivity or late hypoactivity observed (data not shown). Therefore, we decided to present the results for the whole 30 minutes instead. Since the motor hyperactivity in Prp-A53T mice has been reported in other publications, we decided to not perform extra motor behavior tests since it seems that this hyperactivity is a compensatory artifact from the model (Lee et al., 2002; Unger et al., 2006). Since the OFT revealed that Prp-A53T mice were somewhat anxious, we decided to perform the nesting test, which is mostly used to measure depressive-like behavior (Deacon, 2006; Kraeuter et al., 2019) (a symptom observed in Parkinson’s disease, (Barone et al., 2009; Reijnders et al., 2008)). Prp-A53T mice significantly scored lower on this test compared to NC mice. However, trehalose treatment did not reduce this depressive-like behavior.
A few other comments are as follows.
Comment 1. § Why is Parkinson’s disease not mentioned in the title? I am NOT asking to change it, but want clarification because synucleopathy is the main feature of PD.
Reply 1. Some reviewers prefer to specify synucleinopathy model instead of PD model, since the synucleinopathy model does not recapitulate all the clinical features of PD. In the present study, it is relevant to use such progressive model to follow the progression of the disease, compared to a more acute neurotoxin-based model such as MPTP or 6-OHDA models.
Comment 2. § Did you perform a power analysis before deciding on the sample size?
Reply 2. We did not perform a power analysis. However, the sample size was decided based on previous studies from our laboratory, in which 8 animals per group are enough to measure robustly TH expression in the brain and the gut (Thy-1-α-syn and SNCA-OVX animals). The statistical power is good enough since we were able to measure differences in TH expression between the experimental groups.
Comment 3. § Were the animals kept individually or in groups in cages? How the water consumption by each animal was estimated if kept together? The dose should be justified and almost equal for all the animals.
Reply 3. Male mice were kept either alone or in duo in cages to avoid unnecessary fights. The amount of remaining drinking water was measured every 2 days and subtracted to the initial amount of water to calculate the consumption. Drinking water was available ad libitum. When 2 animals were kept together, the initial amount of water was doubled. The remaining amount of water was then divided by 2 to take into account that there were 2 animals per cage. Therefore, we assumed that both animals drank the same amount of water during the experiment, i.e. that the dose of trehalose was the same between animals. Of importance, there was no difference in mice body weight from the same experimental groups, suggesting that no dehydration occurred and that mice drank overall the same amount of water.
Comment 4. § May I know, what was the euthanasia method?
Reply 4. Terminal anesthesia was performed with a ketamine/xylazine mixture (100/10 mg/kg), followed by intracardiac perfusion, as indicated in the methods section.
References
Barone, P., Antonini, A., Colosimo, C., Marconi, R., Morgante, L., Avarello, T.P., Bottacchi, E., Cannas, A., Ceravolo, G., Ceravolo, R., Cicarelli, G., Gaglio, R.M., Giglia, R.M., Iemolo, F., Manfredi, M., Meco, G., Nicoletti, A., Pederzoli, M., Petrone, A., Pisani, A., Pontieri, F.E., Quatrale, R., Ramat, S., Scala, R., Volpe, G., Zappulla, S., Bentivoglio, A.R., Stocchi, F., Trianni, G., Dotto, P.D., PRIAMO study group, 2009. The PRIAMO study: A multicenter assessment of nonmotor symptoms and their impact on quality of life in Parkinson’s disease. Mov Disord 24, 1641–1649. https://doi.org/10.1002/mds.22643
Deacon, R.M., 2006. Assessing nest building in mice. Nat Protoc 1, 1117–1119. https://doi.org/10.1038/nprot.2006.170
Kraeuter, A.-K., Guest, P.C., Sarnyai, Z., 2019. The Nest Building Test in Mice for Assessment of General Well-Being. Methods Mol Biol 1916, 87–91. https://doi.org/10.1007/978-1-4939-8994-2_7
Lee, M.K., Stirling, W., Xu, Y., Xu, X., Qui, D., Mandir, A.S., Dawson, T.M., Copeland, N.G., Jenkins, N.A., Price, D.L., 2002. Human alpha-synuclein-harboring familial Parkinson’s disease-linked Ala-53 --> Thr mutation causes neurodegenerative disease with alpha-synuclein aggregation in transgenic mice. Proc Natl Acad Sci U S A 99, 8968–8973. https://doi.org/10.1073/pnas.132197599
Reijnders, J.S.A.M., Ehrt, U., Weber, W.E.J., Aarsland, D., Leentjens, A.F.G., 2008. A systematic review of prevalence studies of depression in Parkinson’s disease. Mov Disord 23, 183–189; quiz 313. https://doi.org/10.1002/mds.21803
Unger, E.L., Eve, D.J., Perez, X.A., Reichenbach, D.K., Xu, Y., Lee, M.K., Andrews, A.M., 2006. Locomotor hyperactivity and alterations in dopamine neurotransmission are associated with overexpression of A53T mutant human alpha-synuclein in mice. Neurobiol Dis 21, 431–443. https://doi.org/10.1016/j.nbd.2005.08.005

Round 2
Reviewer 2 Report
Comments and Suggestions for Authors
The authors have logically responded to the comments and concerns raised by me. Still, I would recommend performing motor-specific behavioral tests in such kind of future studies.
Comments on the Quality of English LanguageMinor editing would suffice.